# Low Copper Diet—A Therapeutic Option for Wilson Disease?

**DOI:** 10.3390/children9081132

**Published:** 2022-07-28

**Authors:** Ulrike Teufel-Schäfer, Christine Forster, Nikolaus Schaefer

**Affiliations:** Department of General Pediatrics, Adolescent Medicine and Neonatology, Medical Center-University of Freiburg, Faculty of Medicine, University of Freiburg, Mathildenstr. 1, 79106 Freiburg, Germany; christine.forster@uniklinik-freiburg.de (C.F.); nikolaus.schaefer@uniklinik-freiburg.de (N.S.)

**Keywords:** Morbus Wilson, nutrition, copper restriction

## Abstract

Wilson’s disease (WD) is an autosomal recessive inherited disease in which a pathological storage of copper in various organs is the mean pathophysiological mechanism. The therapy consists of drug therapy with chelating agents or zinc. For patients, nutrition is always an important issue. The aim of this review was to determine whether there are clear recommendations for a low copper diet for WD patients, or whether the essential trace element zinc plays a role? We were able to show that some of the foods with high copper content would have to be consumed in such large quantities that this is regularly not the case. Furthermore, there are also different absorption rates depending on the copper content. A lower copper intake only prevents the re-accumulation of copper. In summary, consistent adherence to drug therapy is more important than a strict diet. Only two foods should be consistently avoided: Liver and Shellfish.

## 1. Introduction

Wilson disease (WD) is a rare autosomal recessive disorder with a reported prevalence of 1:30,000–1:50,000 [1,2,3]. In WD, mutations in the adenosine triphosphatase (ATP) 7B gene, which encodes for a copper-transporting ATPase, lead to impaired copper metabolism. The changes in the ATPase cause a reduced excretion of copper via the bile into the intestine and a faulty binding of copper into ceruloplasmin [4], so that increased deposition of copper in liver cell develops; without therapy, this leads to long-term damage and even cirrhosis. Once the diagnosis has been made, drug therapy is initiated [2,5]. The goal of therapy is a consistent removal of the already accumulated copper by chelators (penicillinamine or triethylenetetramine dihydrochloride) and the prevention of a new accumulation, which can be achieved with chelators as well as with zinc. The aim of this review was to determine whether there are clear recommendations for WD patients with regard to a low copper diet. Since zinc is an essential trace element that is also absorbed with food and is a known therapy in WD, we additionally looked for recommendations in this regard.

## 2. Methodology

We searched PubMed without language restrictions from inception to May 2022. Text terms (synonyms and word variations) and database-specific keywords used included WD, diet, nutrition, therapy, medication, and zinc. We also reviewed the reference list of relevant abstracts and original research articles to find other potential studies.

## 3. Copper Metabolism

### 3.1. Pathophysiology of Copper Metabolism

Copper is important for many functions in the body. The fetal copper balance is complicated. ATP7A enables the transport of copper across the placenta to the child. A defect in this transport leads to copper deficiency, the so-called Menkes disease. During pregnancy, Menkes disease results in early embryonic death and fetal structural abnormalities. ATP7B, on the other hand, is responsible for the reverse transport, which creates a balance. As the fetus gets older, the copper requirement also increases so that the function of ATP7B is no longer so important [6]. However, there is no data available on whether children of homozygous mothers with WD already have copper overload due to the defective ATP7B. Studies in murine models have shown that the offspring of sick mothers tend to be copper deficient and do not survive if they are not suckled by healthy dams [7,8].

After food intake, copper is absorbed into the enterocytes, mainly in the duodenum and small intestine, via copper transporter 1 [4]. The average absorption rate lies between 55% and 75% [9]. In enterocytes, transfer into the portal system is regulated by the transmembrane protein ATP7A. A dysfunction of this leads to a copper deficiency (Menkes disease), which means that the copper remains in the enterocytes and is then excreted by the renewal of the intestinal mucosa [10].

After successful uptake into the liver cell, there are three pathways for copper. Firstly, it is bound to proteins and enzymes, thus regulates biological processes, and can be released back into the body’s circulation in small amounts as free non-ceruloplasmin bound copper. Secondly and mainly, copper is structurally bound to ceruloplasmin via ATP7B in the trans-Golgi network. This bound copper can then leave the liver cell and be transported to other organs. On the other hand, copper is excreted into the bile through the formation of vesicles in an ATP7B mediated manner. In WD, there is a malfunction of ATP7B so that copper cannot be bound to ceruloplasmin and also cannot be excreted into the bile. As a result, the copper ion concentration in the cells increases, leading to oxidative stress and subsequent cytotoxic effect; this does not only happen in the liver cells. The increased release of non-ceruloplasmin-bound copper then accumulates in other tissues as well. Oxidative damage of haemoglobin and the cell membrane leads to coombs-negative hemolysis [11]. In addition, more free copper (non-ceruloplasmin bound copper) is released from hepatocytes and circulates in the blood. It is then absorbed and stored in other organs such as the brain and the eye. This results in extrapyramidal disorders, psychiatric illness, dysarthria and Kayser–Fleischer ring in the eye as the main symptoms of WD [4,12].

### 3.2. Copper Absorption

The copper absorption rate has been shown to be dependent on the copper content [13,14]. Overall, more copper is absorbed when there is a high copper content in the diet, but the absorption rate decreases compared to diets with less copper. In a standardized intake, it was shown that the absorption rate varies from 56% with low copper content to 12% with high copper content [14]. Interestingly, with long-term high copper intake, the secretion via the gastrointestinal tract increases. However, the increase is not as high to remove all excess copper [13]. Other factors that affect absorption are age, gender, and diet. Although copper was absorbed less efficiently from a vegetarian diet than from a non-vegetarian diet, the apparent total copper absorption from the vegetarian diet was greater due to its higher copper content [15].

## 4. Discussion

### 4.1. Role of Copper Uptake in WD Therapy

The American and European guidelines usually state that foods with a high copper content (e.g., liver, shellfish, mushrooms, chocolate, dried fruits, and nuts) should be avoided, especially during the first year of treatment [5,16]. These foods are not part of the regular diet of newborns or young children.

In the 6th to 12th month of life, the baby increasingly starts to eat complementary food. The foods used here, e.g., sweet potato, also have a high copper content. With increasing age, the dietary intake of copper in infantile children becomes lower. However, it lies still above the recommended daily requirement, despite avoiding foods with very high copper content.

In order to achieve a too high copper intake, restricted foods would have to be consumed in large amounts. According to European recommendations, an adult should consume around 1.3 mg/day (female) or 1.6 mg/day (man) of copper for an adequate intake [17]. For example, the daily copper requirement is equivalent to 625 g milk chocolate, 440 g mushrooms or 320 g peanut butter. This does not take into account the effectiveness of absorption of about 36% [14].

Moreover, it shows that the amounts of food consumed would have to be unrealistically large. Exceptions are liver and shellfish. Liver contains about 157 mg/kg copper and lobster 36.6 mg/kg. This would exceed the daily requirement for normal consumption, even if the absorption capacity is taken into account.

According to the American guideline, a vegetarian diet can delay the onset of the disease and control its progression by reducing the bioavailability of copper, but should not be used as the sole therapy. Here the statement refers to the case reports of Brewer et al. with two patients who did not adhere to drug therapy but did follow a lacto-vegetarian diet [18]. It should be noted, however, that the pathology of WD is not caused by an excessive intake of copper, but in a hepatocellular excretory disorder. Therefore, lower copper intake can only prevent copper re-accumulation.

### 4.2. What Is the Significance of Zinc as an Essential Trace Element in Therapy?

Copper and the essential trace element zinc have competing functions in the body. Both interact at the level of the intestinal mucosa and influence each other’s absorption. After resorption in the enterocytes, zinc leads to an increased synthesis of metallothionein. This binds copper and prevents its uptake into the portal system. The bound copper is increasingly stored in the enterocytes and excreted through the intestine via the regular renewal of the enterocytes [19]. In addition, it also increases copper-binding metallothioneins in hepatocytes, reducing the deleterious effects of free liver copper.

Unfortunately, we have not found any studies describing the influence of zinc-containing foods with regard to WD. However, zinc has been used in the therapy of WD for a long time. The use in the treatment of patients with symptomatic WD is still controversial [20,21,22]. A present meta-analysis showed that WD patients treated with D-penicillamine and zinc seem to exhibit similar treatment effectiveness in all symptomatic WD patients [22]. D-Penicillamine has also been shown to cause adverse effects and neurological deterioration more frequently. Weiss et al. conducted a retrospective analysis of 288 WD patients in Europe, and the results showed that zinc monotherapy is not as effective as chelators in preventing hepatic deterioration [21].

Therefore, zinc is used primarily in very young children identified by family screening or in adult patients who have already been well detoxified with chelating agents. In the published case reports with infantile WD, additional zinc therapy was usually started in the second year of life in order to avoid copper accumulation in the hepatocytes [23]. However, the exact time to start zinc therapy is uncertain. Furthermore, elevated transaminases also occur at different times as a manifestation of the disease.

Sturm et al. conducted a survey of European and American pediatric hepatologists to find out how they would treat patients in different cases [24]. There were also additional non-case-related questions. For example, one question was whether a low copper diet is recommended for children with newly diagnosed WD and how long it should be followed. Interestingly, there were regional differences. The American colleagues stated that they would recommend a low copper diet. 23% of the European colleagues do not recommend this. The recommendation on duration was also very variable.

## 5. Conclusions

In summary, the complete elimination of copper from the diet is impractical due to its ubiquitous occurrence. There is also no clear evidence for the benefits of a strict low copper diet. Avoiding foods high in copper may be useful after diagnosis and at the beginning of treatment, especially liver and shellfish. In addition, food should not be prepared in pots containing copper. Opinions differ on zinc in already symptomatic MW. Zinc can certainly be used in early childhood after the diagnosis of MW through family screening. However, copper that has already been deposited cannot be removed from the body by avoiding foods containing copper or by zinc. This therapeutic goal can only be achieved by chelation therapy. Therefore, consistent adherence to drug therapy is more important than dietary restriction of copper.

## Data Availability

Not applicable.

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
