# Peer review of "Low Copper Diet—A Therapeutic Option for Wilson Disease?"

_children, 2022, doi:10.3390/children9081132_

Round 1
Reviewer 1 Report
The bibliography is very old, and can only be used for a conceptual framework, not for a review.
They do not provide concise and accurate updates on the latest developments in a given area of research.
Where is the methodology?
Author Response
Response to Reviewers
We thank the editorial team and the reviewers for the opportunity to revise and resubmit the paper. We hope that concerns could be clarified. All revisions to the manuscript have been made using the "Track Changes" feature.
Reviewer 1:
- The bibliography is very old, and can only be used for a conceptual framework, not for a review.
- Thank you for the comment. We again searched the literature for recent studies from the last 5 years.
- They do not provide concise and accurate updates on the latest developments in a given area of Research.
- We are aware that there are recent studies on the topic of Wilson's disease and especially on therapy. In this text, we have focused on nutrition. This aspect receives much less attention in studies. Even in the guidelines, the topic is only briefly touched.
- Where is the methodology?
- We have added a methodology chapter:
- We searched at PubMed without language restrictions from inception to May 2022. Text terms (synonyms and word variations) and database-specific keywords used included wilson disease, diet, nutrition, copper, therapy, medication and zinc. We also reviewed the reference list of relevant abstracts and original research articles to find other potential studies.
Reviewer 2 Report
The article covers an important and insufficiently studied topic. The authors have presented a clear and concise summary of the pathophysiology of Wilson's disease and copper metabolism and have aimed to address the role of diet in the treatment of Wilson's disease.
The topic is interesting and relevant considering the growing interest of patients and families in the role of nutrition in the daily management of chronic liver disease.
The main part of the article, which is the available evidence and recommendations regarding low-copper diet in Wilson's disease merits, in my opinion, a more thorough discussion. The authors have decided to concentrate on the role of low-copper diet rather than discuss the overall nutrition in Wilson's disease, which is a valid choice, however they also include a discussion of the role of zinc, which would necessitate rewriting or reconsidering of the stated aims of the article.
The chapter "Discussion Copper and zinc uptake in infancy and adults" needs, in my opinion, to be restructured. In its current form, it's fairly short and the discussion of zinc is interspersed with other aspects of the diet. It would be more readable if discussion of the role of zinc were moved to a separate subchapter and the physiology or zinc absorption as well as the theoretical role of zinc were discussed *before* the therapeutic recommendations were introduced.
The title "Discussion Copper and zinc uptake in infancy and adults" needs rewording as it is grammatically incorrect.
The sentence in line 82
"breast milk and formula food also has a high copper content. It contains > 150% of the daily requirement."
is unclear. The coverage of daily requirement depends, among other things, on the specific formula, and the amount ingested, and so the sentence needs a lot of context. The copper content in breast milk and formula varies widely (as cited in [17]) so a more thorough discussion, and possibly inclusion of more examples, of copper content in different food would be helpful - possibly in the form of a table.
The reasoning in the lines 107 to 115 is not entirely justified. The argument seems to be that since it is difficult to exceed recommended copper intake by consuming high-copper foods since it would necessitate ingesting large amounts, therefore copper content in the diet of WD patient is not really relevant. This argument omits the fact that Wilson's disease is primarily caused by disturbance of copper elimination rather than excessive intake, and so the conclusion is oversimplified.
In conclusion, I think this valuable article would benefit from expansion of its more important chapter.
Author Response
Response to Reviewers
We thank the editorial team and the reviewers for the opportunity to revise and resubmit the paper. We hope that concerns could be clarified. All revisions to the manuscript have been made using the "Track Changes" feature.
Reviewer 2:
The article covers an important and insufficiently studied topic. The authors have presented a clear and concise summary of the pathophysiology of Wilson's disease and copper metabolism and have aimed to address the role of diet in the treatment of Wilson's disease.
The topic is interesting and relevant considering the growing interest of patients and families in the role of nutrition in the daily management of chronic liver disease.
- The main part of the article, which is the available evidence and recommendations regarding low-copper diet in Wilson's disease merits, in my opinion, a more thorough discussion. The authors have decided to concentrate on the role of low-copper diet rather than discuss the overall nutrition in Wilson's disease, which is a valid choice, however they also include a discussion of the role of zinc, which would necessitate rewriting or reconsidering of the stated aims of the article.
- We thank you for the comment and we have revised this.
- The chapter "Discussion Copper and zinc uptake in infancy and adults" needs, in my opinion, to be restructured. In its current form, it's fairly short and the discussion of zinc is interspersed with other aspects of the diet. It would be more readable if discussion of the role of zinc were moved to a separate subchapter and the physiology or zinc absorption as well as the theoretical role of zinc were discussed *before* the therapeutic recommendations were introduced.
- We completely agree with the reviewer and have revised the section.
- The title "Discussion Copper and zinc uptake in infancy and adults" needs rewording as it is grammatically incorrect.
- The following amendment has been made:
- 4.Discussion
- 4.1 Role of copper uptake in WD therapy
- 4.2 What is the significance of zinc as an essential trace element in therapy?
- The sentence in line 82 "breast milk and formula food also has a high copper content. It contains > 150% of the daily requirement." is unclear. The coverage of daily requirement depends, among other things, on the specific formula, and the amount ingested, and so the sentence needs a lot of context. The copper content in breast milk and formula varies widely (as cited in [17]) so a more thorough discussion, and possibly inclusion of more examples, of copper content in different food would be helpful - possibly in the form of a table.
- We have deleted the passage regarding breast milk, formulas, and copper content. We had based the text on Supplemental Table 2 by Valentino et al (Supplemental Digital Content, http://links.lww.com/MPG/B762; Journal of Pediatric Gastroenterology & Nutrition. 2020;70(5):547-54). At your request to provide a table on this, we did our own calculations. According to the DACH reference values of the German Society for Nutrition, the recommended amount of copper in e.g. 4th-12th months of life is 0.6-0.7mg/day. In Neocate, the standard solution contains 0.057 mg copper / 100 ml. This means that with a drinking quantity of 800ml per day only 0.46 mg copper is drunk. Whereby this amount is not completely absorbed. Thus, one does not come above the recommended intake. We have also calculated this for other formulas such as Aptamil 1, Nutramigen and Beba. With none of the foods comes relevant times over the recommended intake.
- We apologize for not having calculated this beforehand.
- The reasoning in the lines 107 to 115 is not entirely justified. The argument seems to be that since it is difficult to exceed recommended copper intake by consuming high-copper foods since it would necessitate ingesting large amounts, therefore copper content in the diet of WD patient is not really relevant. This argument omits the fact that Wilson's disease is primarily caused by disturbance of copper elimination rather than excessive intake, and so the conclusion is oversimplified.
- We completely agree with the reviewer and the following amendment has been made:
- It should be noted, however, that the pathology of wilson disease is not caused by an excessive intake of copper, but in a hepatocellular excretory disorder. Therefore, lower copper intake can only prevent copper re-accumulation.
In conclusion, I think this valuable article would benefit from expansion of its more important chapter.